# Development and Validation of Oral Health Knowledge, Attitude and Behavior Questionnaire among Indian Adults

**DOI:** 10.3390/medicina58010068

**Published:** 2022-01-02

**Authors:** Siddharthan Selvaraj, Nyi Nyi Naing, Nadiah Wan-Arfah, Mohmed Isaqali Karobari, Anand Marya, Somasundaram Prasadh

**Affiliations:** 1Faculty of Medicine, Medical Campus, Universiti Sultan Zainal Abidin, Kuala Terengganu 20400, Terengganu, Malaysia; sidzcristiano@gmail.com; 2Faculty of Health Sciences, Universiti Sultan Zainal Abidin, Kuala Terengganu 20400, Terengganu, Malaysia; wanwaj@unisza.edu.my; 3Conservative Dentistry Unit, School of Dental Sciences, Universiti Sains Malaysia, Health Campus, Kubang Kerian, Kota Bharu 16150, Kelantan, Malaysia; dr.isaq@gmail.com; 4Department of Restorative Dentistry & Endodontics, Faculty of Dentistry, University of Puthisastra, Phnom Penh 12211, Cambodia; 5Department of Orthodontics, Faculty of Dentistry, University of Puthisastra, Phnom Penh 12211, Cambodia; amarya@puthisastra.edu.kh; 6Department of Orthodontics, Saveetha Institute of Medical and Technical Sciences, Saveetha Dental College, Saveetha University, Chennai 600077, India; 7ACP Office—Research, National Dental Centre Singapore (NDCS), Singapore 168938, Singapore

**Keywords:** psychometric properties, reliability, oral health, knowledge, attitude and behavior

## Abstract

*Background and objectives*: The Indian population faces numerous challenges to attain better oral hygiene due to a lack of oral health literacy. For the past 10 years, the prevalence of dental-related conditions in India has become a considerable problem in every state of India. A health-education-based oral health promotion strategy will be an ideal choice for the Indian population instead of endorsing conventional oral health promotion. The use of unsuitable tools to measure may lead to misleading and vague findings that might result in a flawed plan for cessation programs and deceitful effectiveness. Therefore, the research aimed to develop and validate an instrument that can assess the oral health knowledge, attitude and behavior (KAB) of adults in India. *Materials and Methods*: This study was carried among adults in India, who live in Chennai, Tamil Nadu. A questionnaire was fabricated and then validated using content, face, as well as construct. The knowledge domain was validated using item response theory analysis (IRT), whereas exploratory factor analysis (EFA) was used to validate the behavior domain and attitude. *Results*: Four principal sections, i.e., knowledge, attitude, demography and behavior, were used to fabricate a questionnaire following validation. Following analysis of item response theory on the knowledge domain, all analyzed items in the domain were within the ideal range of difficulty and discrimination. The Kaiser–Meyer–Olkin measure of sampling adequacy was 0.65 for the attitude and 0.66 for the behavior domain. A Bartlett’s test of sphericity was conducted and demonstrated that outcomes for both domains were highly significant (*p* < 0.001). The factor analysis resulted in three factors with a total of eight items in the attitude domain and three factors with a total of seven items in the behavior domain depicting satisfactory factor loading (>0.3). Across the three factors, i.e., knowledge, attitude and behavior, internal consistency reliability was tested using Cronbach’s alpha, and the values obtained were 0.67, 0.87, 0.67, and 0.88, respectively. *Conclusions*: The findings of this study that assessed validity and reliability showed that the developed questionnaire had an acceptable psychometric property for measuring oral health KAB among adults in India.

## 1. Background of the Study

Oral health disorders are considered to have a widespread presence, which is associated with remarkable morbidity. The existence of dental caries is widespread around the globe, and, on the other hand, 15% of the adult population considerably experience severe periodontal disease [1]. The Indian population has a significant imbalance in oral health care compared to general health [2,3]. Oral health-related conditions can be prevented with proper and effective home oral hygiene measures [4,5]. Proper maintenance of oral hygiene will have a better impact on an individual’s general health [6]. For the past 10 years, the prevalence of dental-related conditions in India has become more common in every state of India [7]. Oral health literacy is considered a factor in deciding the outcome of oral health policies and programs [8]. A health-education-based oral health promotion strategy is an ideal choice for India, instead of endorsing conventional oral health promotion methods, which are unsuccessful in attaining the alteration that is followed in developed nations [9,10]. Researching oral health knowledge, attitude and behavior would be an ideal approach to enhance adolescent oral health literacy who do not have the chance to attain oral health literacy during adulthood due to the lack of importance given to oral health [11]. Several studies have stated that an individual’s knowledge and attitude are interrelated with control and illness prevention [12], and medical treatment response, such as dental implant procedures, that would enhance the quality of life of an individual [13,14]. Assessing the level of knowledge, attitude and behavior of an individual by using a questionnaire is ideal during the ongoing pandemic all over the world [15], thus we developed our very own questionnaire to satisfy the required research outcome instead of adopting an existing questionnaire.

Reliability with precision manifests the range to which the measurement tool reproduces by determining its internal consistency [16]. The standard primary method to evaluate KAB is by questionnaires [17]. There are questionnaires currently available to estimate oral health knowledge, attitude and behavior, but we developed our own version of the questionnaire to maintain the flow, understandability, and contents to achieve the objectives of our study and make our research a productive one. Our questionnaire can also be used as an ideal tool to assess the oral health knowledge, attitude and behavior among adults as the content in our questionnaire satisfies psychometric properties [18]. This study is carried out to develop and validate an oral health KAB instrument that can assess face validity, content validity, construct validity, and internal consistency reliability towards oral health among adults in India that helps to enhance the oral hygiene of an individual.

## 2. Materials and Methods

### 2.1. Research Design and Study Population

This cross-sectional study was conducted between the month of December 2020 to January 2021 among adults who live in Tamil Nadu, India. The validation of the questionnaire took place among residents of Chennai, Tamil Nadu.

### 2.2. Sample Size and Sampling Method

This study was carried out among 225 adults from Chennai, Tamil Nadu. Adults who were above 18 years and volunteered to participate in this study were included. Physically and mentally incapacitated individuals were not included in this study as consent cannot be obtained. Simple random sampling was applied to select the participants. A brief explanation was given about the study and the study outcome to the participants. As the world is facing a pandemic situation due to the COVID-19 outbreak, strict SOP were followed during the data collection procedure [19]. After obtaining consent from the participants, the questionnaires were distributed and returned once the participants filled it. By rule of thumb, based on the recommendation by Klein in the year 2011, the sample size was calculated to validate the questionnaire by exploratory factor analysis for construct validity [20].

### 2.3. Questionnaire Development

The questionnaire and factor analysis development were carried out in two stages: stage one is questionnaire development, followed by psychometric evaluation. The second stage comprises three analyses: (i) exploratory factor analysis (EFA) for construct validity, (ii) item response theory (IRT) analysis, and (iii) internal consistency reliability analysis.

### 2.4. Stage One: Items and Domains Development

During stage one, a vast search was conducted on the literature to obtain resources on oral health knowledge, attitude and behavior and find the applicable scales and items on prevailing questionnaires. Ethical approvals were obtained from the UniSZA Human Research Ethics Committee (ref no: UniSZA/UHREC/2020/197) and RIPON independent ethics committee (ref no: RIPON/NOV30/2020/800). The questionnaire was developed to be anonymous. The data obtained in this study were kept confidential. Extensive interviews were carried out on adults of Tamil Nadu to survey qualitatively on knowledge, attitude and behavior towards oral health. An evaluation guide was developed and used during the interview, and it comprises of 58 items that were found in the extensive review of literature covering demography, knowledge, attitude and behavior domains. The interview conducted among the adults towards oral health was replicated and analyzed by content analysis. The initial questionnaire draft was reviewed by an expert panel that comprised dentists, educationists, nurses, and statisticians to validate the contents of the questionnaire with the intended theories and constructs.

The interpretations obtained from the interviews on the level of knowledge among the responders were used to produce the appropriate questionnaire domains. The knowledge domain was developed based on risk factors, etiology, complications of oral-related diseases and symptoms. Secondly, the attitude domain of the questionnaire was developed based on Health Belief Model (HBM) theory [21]. On the other hand, questions in the behavior domain were developed based on the oral-health-related conditions preventive strategies provided by the World Health Organization and the Centre for Disease Control and Prevention.

Content validity of the oral health KAB questionnaire was carried out with an expert panel that included a dentist, epidemiologist, statistician and a health educator. The expert team selected the ideal items for question precision, knowledge accuracy, attitude and behavior interpretability. This expert team also helped to find and judge the content validity of the items selected initially for questionnaire inclusion. The questionnaire was developed in the English language. Face validity was carried out among 10 adults in Chennai to assess the layman’s understanding of the questionnaire and know how the items are meaningful to the target population [22].

The participants were requested to explain and assess each item that was present in the questionnaire after the open-ended discussion. Based on their understanding and answers given to the question, an assessment of vagueness was carried out. The final version of the questionnaire was established from the face validation findings. The self-administered questionnaire comprised of open and close ended questions, which was received well by the participants in the study. The questionnaire has 4 domains in the final questionnaire consisting of 39 questions; the domains in the questionnaire were (1) demography of participants; (2) knowledge towards oral health; (3) attitude towards oral health; and (4) behavior towards oral health. The socio-demographic characteristics that were studied in the study included age, gender, race, religion, diet, smoking habit, alcohol habit, marital status, occupational status, level of education, income, ownership of a house and ownership of a vehicle. The domains, questions and response choices in the questionnaire are shown in Table 1.

### 2.5. Second Stage: Validation

The data collection was carried between December 2020 to January 2021. A total of 225 participants were included from Chennai to assess the psychometric property of the questionnaire. At first, the study participants were given a brief explanation of the study, and informed consent was obtained from the respondents participated in the study. Oral health KAB questionnaires were distributed to the individuals for self-administration. R software version 3.6 was used to analyze the data. A value of 0.05 was set as the significance level.

### 2.6. Item Response Theory (IRT)

The sample size needed for 2-PL item response theory is not well specified; however, some studies have suggested a range of samples between 100 and 500 [23]. The sample size around 225 participants was deemed acceptable for IRT analysis of the knowledge domain. Two-parameter logistic item response theory (2-PL IRT) analysis was carried out for the knowledge domain with responses in dichotomous output as either correct answer or wrong answer. The analysis was performed in R software (version 3.6) using the RASCH function of the ltm package. An acceptable range of difficulty (−4 to +4) and discrimination (0.20 to infinity) was considered the cut-off value for the psychometric properties’ evaluation of the domain. Item fit was evaluated by the Chi-square goodness-of-fit per item and represented with corresponding *p* values, and one-dimensionality was analyzed by modified parallel analysis.

### 2.7. Exploratory Factor Analysis (EFA)

According to a study by Kline in 2011, the desired sample size to conduct EFA is between 2 and 5 participants per question [20]; subsequently, based on the total number of items (45) in the current study, the sample size decided upon was 225. The EFA was carried out to determine the construct validity of the questionnaire’s knowledge attitude and behavior domains. Sampling adequacy was determined using Kaiser–Meyer–Olkin measure (KMO) and Bartlett’s test of the sphericity [24]. The sample was regarded sufficient if the KMO value was above 0.5 and Bartlett’s test was significant (*p* < 0.001). For the component extraction, principal axis factoring method was used. Components with eigenvalues of over one were retained based on the Kaiser–Guttman rule [25]. Oblimin rotation along with Kaiser normalization was used to optimize the loading factor for selected components. Items with a loading factor of above plus or minus 0.5 were found to be acceptable loading factors [26].

### 2.8. Internal Consistency Reliability

A coefficient of Cronbach’s alpha calculated the internal consistency (IC) of the items. The items in the questionnaire were found to represent good internal consistency [27]. In this study, the items’ internal consistency (IC) was calculated using Cronbach’s alpha coefficient and correlation between the items.

## 3. Results

Initially, 58 questions in all 4 sections of the questionnaires were formatted. Demography, knowledge, attitude and behavior are the four areas considered. The demographic domain contained 13 questions, but each of the remaining domains contained 15 questions. For descriptive analysis and to research the socio-demographic characteristics of the 225 participants, data from the demographic domain were used. For knowledge, attitude and behavior domains, exploratory factor analysis (EFA) was carried out. The final draft of the questionnaire consisted of 4 domains and 39 items (13 items on demography, 11 items on knowledge, 8 items on attitude and 7 items on behavior).

### 3.1. Questionnaire Development, Content Validity and Face Validity

Literature studies was carried out extensively on oral health and concepts that were useful for our study, and ideas were taken in account in which essential items are generated for the domains that might fit in the questionnaire. Once the expert panel reviewed the questionnaire extensively, the items were incorporated in the domains with appropriate consistency. An elaborate interview was carried out to develop suitable domains and incorporation of items in the questionnaire. During this procedure, the required items can be incorporated into the questionnaire based on the individual’s oral-health-related responses. Later, the pretesting of the questionnaire was carried out on the next session among the adults to determine the face validity of the questionnaire. Depending on the participants’ opinions, the significant words and terminologies used in the questionnaire were straightforward and clear to understand. Still, there were a few confusing words, and terminologies were altered for better understanding. Overall, the participants did not feel any difficulty understanding the items in the questionnaire.

### 3.2. Socio-Demographic Characteristics

For all 225 people who participated in the research, descriptive statistics were carried out. Table 2 summarizes the socio-demographic characteristics of the participants. The study included 133 (59%) males and 92 (41%) females. The age groups of 18–24 years, 25–34 years, 35–44 years and ≥45 years were 63 (28%), 77 (34%), 6 (25%) and 29 (13%) participants, respectively. A total of 158 (70%) out of 225 participants were married. Hinduism, Islam, Christianity and other religions were followed by 115 (51%), 76 (34%), 26 (12%) and 8 (4%) members, respectively. The majority of 209 (93%) participants belonged to the Tamil ethnicity, and the remaining 16 (7%) belonged to a non-Tamil ethnicity. A mixed form of diet was eaten by most of the members (60%). Most of the participants in the study (70%) were non-smokers (78%) and non-alcoholic. In terms of educational history, there was a total of 3 (1%) illiterates, 35 (16%) with primary school education, 90 (40%) with secondary school education and 97 (43%) with university-level education among the study participants. In this survey, 138 (61%), 13 (6%), 61 (27%) and 13 (6%) respondents were, respectively, working, unemployed, students and homemakers. A total of 86 (38%) participants had an income below ₹10 K, 13 (6%) received an income of ₹10 K–₹20 K, 61 (27%) received an income of ₹20 K–₹30 K and 65 (29%) received an income above ₹30 K. Nearly 125 (56%) of them lived in a leased home, while 100 (44%) lived in a rented house. Almost 121 (54%) participants did not own a vehicle, while 104 (46%) owned a vehicle.

### 3.3. IRT for Knowledge-Based Questions

The knowledge domain was analyzed by item response theory; the psychometric properties of the knowledge domain are shown in Table 3. Items 12, 15, 3 and 4 in the questionnaire were not considered due to poor psychometric properties. All items within the range of −4 to +4 difficulty parameters were retained in the questionnaire. Items K1, K2, K5, K6, K7, K8, K9, K10, K11, K13 and K14 were retained. For all the retained items, values for discrimination were found to be in the range of 0.21 to 3.12, as shown in Table 3. The goodness-of-fit showed that none of the items fitted well (*p* > 0.05). All the items in the questionnaire were retained with expert advice because of the importance and relevance in determining the knowledge of the participants. The amount of information tapped by the items between −4 and +4 difficulty was determined to be 85%. Knowledge domain consists of items K1 (”There are two sets of teeth during lifetime”), K2 (“Tooth infection causes gum bleeding”), K5 (“Replacement of a missing tooth improves oral hygiene”), K6 (“The dental caries of deciduous teeth need not be treated”), K7 (“Bacteria are one of the reasons to cause gingival problem”), K8 (“Fizzy soft drinks affect the teeth adversely”), K9 (“Loss of teeth can interfere with speech”), K10 (“Irregularly placed teeth can be moved into the correct position by dental treatment”), K11 (“Decayed teeth can affect the appearance of a person”), K13 (“Tobacco chewing, or smoking can cause oral cancer”) and K14 (“White patches on teeth are called dental plaque”). Discrimination >1 and difficulty between −4 to 4 were accepted. Overall, Cronbach’s alpha was 0.67 (95% CI). A total of 95% of information is tapped.

### 3.4. EFA for Attitude-Based Questions

A total of 15 questions in the attitude parameter were considered for EFA and were accompanied by a KMO (Kaiser–Meyer–Olkin) sampling adequacy measure of 0.65 and a Bartlett’s sphericity test significance of <0.001. The five-factor model was proposed by the principal component analysis (PCA), but, based on the greater eigenvalue criterion, three factors were defined for the implementation of EFA. Then, these three factors were rotated using oblimin rotation. The selection of questions was determined based on the loading factors where the value would be >0.5. EFA retained questions (items) 12, 2 14, 10, 11, 8, 13 and 3 in the attitude domain (Table 4).

### 3.5. EFA for Behavior-Based Questions

A total of 15 questions of behavior domain were considered for EFA with KMO (Kaiser–Meyer–Olkin) measure of sampling adequacy of 0.66 and significance of Bartlett’s test of sphericity <0.001. EFA was deliberated with a three-factor model based on the greater eigenvalue criterion and these factors were further rotated using oblimin rotation. Questions for the factors were maintained with a value greater than 0.5 for loading factors. Questions (items) 6, 12, 9, 10, 13, 1 and 5 were retained by EFA (Table 5).

### 3.6. Internal Consistency Reliability

The Cronbach’s alpha of the overall questionnaire was 0.67, 0.87 and 0.88 for knowledge, attitude and behavior domains, respectively, which showed acceptable internal consistency reliability. For the attitude section, the three factors (daily oral hygiene, oral hygiene habits, and oral hygiene assumptions) showed ideal internal consistency of 0.90, 0.86 and 0.85, respectively. Likewise, the behavior domain has an ideal Cronbach’s alpha value of 0.93, 0.94 and 0.81 for three factors (behavior towards teeth, behavior towards teeth health and behavior towards teeth condition).

## 4. Discussion

To our best knowledge, this is the first study that has developed and validated a questionnaire that has satisfactory content, face validity and reliability for the examination of the oral health knowledge, attitude and behavior of adults in Chennai.

The oral health knowledge, attitude and behavior questionnaire construction was conducted in the present study. Validation of the questionnaire was achieved by using item response theory and exploratory factor analysis. Some items were omitted based on the validation and assessment of the questionnaire as they were found to be uncertain. Some were deleted during the content validity evaluation and some due to the problematic validity related to assessment of KAB towards oral health. Results obtained from exploratory factor analyses showed an ideal structure for the new tool developed.

The accepted psychometric property towards the knowledge domain was found by item response theory analysis. The absolute discrimination value range was between minus infinity to plus infinity; however, questions that exhibited negative figures of discrimination were found to be problematic as they infer that the participants with a high score are less expected to keep up a firmer response alternative [28].

All the retained items showed discrimination parameters to be positive and easy [29]. All factor loadings were more than 0.3, which shows a close association among factors and items [30], and questions in the knowledge section showed an overall Cronbach’s alpha value of 0.67. Items K13 (“Tobacco chewing, or smoking can cause oral cancer”), K8 (“Fizzy soft drinks affect the teeth adversely”), K11 (“Decayed teeth can affect the appearance of a person”), K2 (“Tooth infection causes gum bleeding”), K14 (“White patches on teeth are called dental plaque”), K10 (“Irregularly placed teeth can be moved into the correct position by dental treatment”), K1 (“There are two sets of teeth during lifetime”), K9 (“Loss of teeth can interfere with speech”), K7 (“Bacteria are one of the reasons to cause gingival problems”), K5 (“Replacement of missing tooth improves oral hygiene”), K6 (“The dental caries of deciduous teeth need not be treated”), based on the value obtained the knowledge domain, made more sense to the questionnaire.

For the attitude domain, an exploratory factor analysis indicated that the three-factor structure of the questionnaire and reliability analysis of the attitude domain exhibited a satisfactory overall Cronbach’s alpha value of 0.87. The retained items, A12 (“Brushing teeth twice a day improves oral hygiene”), A2 (“Keeping your teeth clean and healthy is beneficial to your health”), A14 (“Improper brushing leads to gum disease”), A10 (“Sweets retention leads to tooth decay”), A11 (“Brushing with fluoridated toothpaste prevent tooth decay”), A8 (“Dentists care only about treatment and not prevention”), A13 (“Gum bleeding denotes gum infection”) and A3 (“Scaling is harmful for gums”), in attitude domain made the questionnaire meaningful by eliminating redundant items. The Cronbach’s alpha value was found to be ideal. The correlation of each item in the attitude domain was more than 0.50, indicating the inter-relatedness of items, and all the items with a loading factor of more than 0.50 were retained [31]. In comparison, Cronbach’s alpha value was higher for attitude and indicated better internal consistency within the attitude domain [32].

For the behavior questionnaires, a three-factor EFA model obtained the overall Cronbach’s alpha value of 0.88. Behavior domain was validated by retaining the following items: B6 (“I give importance to my teeth as much as any part of my body”), B12 (“I have sensitive teeth”), B9 (“I brush my tooth twice daily”), B10 (“I use teeth to open the cap of bottled drinks”), B13 (“I experience toothache while chewing food”), B1(“I have bleeding gums during brushing”) and B5 (“I do routine dental check-up”). The Cronbach’s alpha value was highest among all domains, indicating a better internal consistency, and a higher correlation within the behavior domain would show ideal precision of evaluation and assessment [33].

This study’s limitation is that the IRT and EFA were carried out to assess validity and reliability. Nevertheless, it is advised that CFA (confirmatory factor analysis) must be performed for validation of the oral health knowledge, attitude and behavior questionnaire.

## 5. Conclusions

A questionnaire should be ideal, easy, simple, in a logical manner and should be in easy to understand, hence we developed our own questionnaire to make it supreme to assess oral health KAB among Indian adults. In this study, a newly developed and validated questionnaire to determine oral health knowledge, attitude and behavior was developed among samples from India. The final version of the questionnaire comprised 4 sections and 39 items (13 items included in demographic characteristics, 11 items included in the knowledge domain, 8 items in attitude and 7 items included in the behavior domain). The oral health KAB domain collectively comprised 26 items, which was ideal to analyze and examine an individual’s oral health KAB level. Although there are several oral health KAB questionnaires available, our developed questionnaire would be ideal to assess the oral health KAB among adults in India. Based on our study findings, the knowledge, attitude and behavior domains possessed acceptable psychometric properties with good construct validity and reliability results that would reflect the participants oral health knowledge, attitude and behavior. The developed questionnaire would be helpful for future studies that are carried out to assess oral health KAB among adults as it satisfies the psychometric property with acceptable construct validity and reliability outcomes.

## Figures and Tables

**Table 1 medicina-58-00068-t001:** KAB Questionnaire towards Oral Health.

Domains	Total Items	Measurement	Response Choice
**Demography**	13	Socio-demographic characteristics: age, gender, education, income, ethnicity, occupation, etc.	Open ended, closed ended, multiple choice
**Knowledge**	11	Etiology, clinical manifestation, treatment, symptoms, preventive measures on oral health	Yes/No/I don’t know. 1 = correct answer. 0 = wrong/I don’t know
**Attitude**	8	Individuals attitude towards oral health based on health belief model	1 = Strongly agree 2 = Agree 3 = Neither agree nor disagree 4 = Disagree 5 = Strongly disagree
**Behavior**	7	Various action towards oral hygiene that might have a good or ill effect on oral health	1 = Never 2 = Seldom 3 = Occasional 4 = Very often 5 = Always

**Table 2 medicina-58-00068-t002:** Socio-Demographic Characteristics of Study Population (*n* = 225).

Parameter		*n*	%
**Age**	18–24 years	63	28.00
	25–34 years	77	34.22
	35–44 years	56	24.89
	≥45 years	29	12.89
**Gender**	Male	133	59.11
	Female	92	40.89
**Marital Status**	Yes	158	70.22
	No	67	29.78
**Religion**	Hindu	115	51.11
	Muslim	76	33.78
	Christian	26	11.56
	Others	8	3.56
**Ethnicity**	Tamil	209	92.89
	Others	16	7.11
**Diet**	Vegetarian	88	39.11
	Non-vegetarian	2	0.89
	Mixed	135	60.00
**Smoking**	Yes	50	22.22
	No	175	77.78
**Alcohol**	Yes	67	29.78
	No	158	70.22
**Education**	Illiterate	3	1.33
	Primary	35	15.56
	High school	90	40.00
	University	97	43.11
**Employment**	Employed	138	61.33
	Unemployed	13	5.78
	Student	61	27.11
	Homemaker	13	5.78
**Income**	Below 10 K ₹	86	38.22
	10 K ₹–20 K ₹	13	5.78
	20 K ₹–30 K ₹	61	27.11
	Above 30 K ₹	65	28.89
**House**	Owned	125	55.56
	Rented	100	44.44
**Vehicle**	Yes	104	46.22
	No	121	53.78

**Table 3 medicina-58-00068-t003:** Results of IRT for knowledge-based questions.

Items	Difficulty	Discrimination	χ^2^ (df = 14)	*p* Value
K4: Periodontal health does not have any effect on diabetic patients	−19.16	−0.07	5.14	0.74
K15: Dental plaque leads to oral ulcer	−5.70	−0.31	3.43	0.90
K12: Improper brushing can cause tooth decay	−2.06	0.64	5.14	0.74
K2: Tooth infection causes gum bleeding	−1.76	1.16	17.43	0.03
K7: Bacteria are one of the reasons to cause gingival problems	−1.73	1.09	13.33	0.10
K14: White patches on teeth are called dental plaque	−1.69	1.04	20.54	0.008
K13: Tobacco chewing, or smoking can cause oral cancer	−1.57	1.23	14.78	0.06
K9: Loss of teeth can interfere with speech	−1.44	1.25	22.40	≤0.001
K6: The dental caries of deciduous teeth need not be treated	−1.43	1.19	6.20	0.62
K8: Fizzy soft drinks affect the teeth adversely	−1.33	1.49	14.37	0.07
K11: Decayed teeth can affect the appearance of a person	−1.29	2.20	11.49	0.18
K10: Irregularly placed teeth can be moved into the correct position by dental treatment	−1.12	2.24	26.23	≤0.001
K1: There are two sets of teeth during lifetime	−1.04	3.12	8.35	0.40
K5: Replacement of a missing tooth improves oral hygiene	−0.83	1.10	22.07	≤0.001
K3: Fluorides prevents tooth decay	5.95	0.21	11.02	0.20

**Table 4 medicina-58-00068-t004:** Result of EFA for attitude-based questions.

Factors	Items	Factor Loading	Correlation	α if Term Deleted	Cronbach’s α
Daily Oral Hygiene	12. Brushing teeth twice a day improves oral hygiene	0.90	0.87	0.89	0.90
	2. Keeping your teeth clean and healthy is beneficial to your health.	0.91	0.87	0.76	
Oral Hygiene Habits	14. Improper brushing leads to gum disease	0.87	0.82	0.72	0.86
	10. Sweets retention leads to tooth decay	0.91	0.82	0.80	
Oral Hygiene assumptions	11. Brushing with fluoridated toothpaste prevents tooth decay	0.77	0.71	0.83	0.85
	8. Dentists care only about treatment and not prevention	0.78	0.69	0.84	
	13. Gum bleeding denotes gum infection	0.78	0.84	0.74	
	3. Scaling is harmful for gums	0.93	0.41	0.94	

**Table 5 medicina-58-00068-t005:** Result of EFA for behavior-based questions.

Factors	Items	Factor Loading	Correlation	α If Term Deleted	Cronbach’s α
Behavior towards teeth	6. I give importance to my teeth as much as any part of my body	0.94	0.94	0.85	0.93
	12. I have sensitive teeth	0.88	0.86	0.92	
	9. I brush my teeeth twice daily	0.89	0.87	0.92	
Behavior towards teeth health	10. I use my teeth to open the cap of bottled drinks	−0.90	0.77	0.90	0.94
	13. I experience tooth ache while chewing food	−0.91	0.77	0.86	
Behavior towards teeth conditions	1. I have bleeding gums during brushing	0.77	0.76	0.67	0.81
	5. I do routine dental check-up	0.82	0.76	0.70	

## Data Availability

The corresponding author will provide the dataset of this study upon request.

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
