# Peer review of "Development and Validation of Oral Health Knowledge, Attitude and Behavior Questionnaire among Indian Adults"

_medicina, 2022, doi:10.3390/medicina58010068_

Round 1

Reviewer 1 Report

The article entitled “Development and Validation of Oral Health Knowledge, Attitude and Behaviour Questionnaire among Indian Adults” aimed to develop and validate an instrument that can assess the knowledge, attitude, and behavior (KAB) towards oral health among Indian adults and the findings of this validity and reliability study showed that the developed questionnaire had a satisfactory psychometric property for measuring KAB towards oral health among Indian adults.

 Authors have well revised several issues; however, I ask authors to add some key concepts. 

  • The introduction section is too short, authors must discuss other tools previously described and may stress the important role that questionnaires (and / or telemedicine) played during the Sars-CoV-2 pandemic (please consider: https://doi.org/10.3390/ijerph17165780),
  • Why haven't the authors discussed and asked questions regarding dental implants? It would be interesting to evaluate the perception of implant survival rates among patients also given the increasingly growing problems related to marginal bone loss (please see Sinjari B, D'Addazio G, Traini T, et al. A 10-year retrospective comparative human study on screw-retained versus cemented dental implant abutments. J Biol Regul Homeost Agents. 2019;33(3):787-797 and 10.1177/03946320110240S214)
  • The bibliography needs to be formatted, please review the journal instructions for authors
  • The conclusion should be more supported by the results

Author Response

See attachement.

Reviewer 2 Report

First of all, thank you for the opportunity to review this manuscript.

The following are suggestions for the present manuscript:

1) Authors should better define the keywords used in the manuscript.

2) The introduction is short and vague. The authors should expand the introduction. The authors need to expand the number of references used in the introduction. I advise them to add recent bibliographical references, from 2020 or 2021.

3) In Material and Method the authors should indicate in which year the study was carried out.

4) What were the inclusion and exclusion criteria of the study?

5) Was only one previous study taken into consideration to calculate the sample size?

6) Reference 16 in the manuscript does not match the bibliographic reference at the end of the manuscript.

7) Have the authors analysed whether there are statistically significant differences in the age and gender of the study participants?

8) How was the employment and education of the respondents analysed?

9) In the discussion the authors do not comprehensively compare their results with results published by other authors. Please discuss.

10) The authors should check the bibliographical references. Bibliographical references are not described according to the journal's guidelines.

Author Response

See Attachement

Round 2

Reviewer 1 Report

The authors adequately addressed the recommendations of this reviewer

Reviewer 2 Report

The reviewer would like to thank the authors for their revision.